# RLAC: Reinforcement Learning with Adversarial Critic for Free-Form Generation Tasks

**Mian Wu**[†1] **Gavin Zhang**[2] **Sewon Min**[2] **Sergey Levine**[2] **Aviral Kumar**[3]
[1]Shanghai Jiao Tong University   [2]UC Berkeley   [3]Carnegie Mellon University
†Work done when visiting UC Berkeley
nothern42@sjtu.edu.cn
**Project website:** https://mianwu01.github.io/RLAC-website/

## Abstract

Open-ended generation tasks require outputs to satisfy diverse and often implicit task-specific evaluation rubrics. The sheer number of relevant rubrics leads to prohibitively high verification costs and incomplete assessments of a response, making reinforcement learning (RL) post-training with rubric-based rewards difficult to scale. This problem is exacerbated by the fact that often the best way to combine these rubrics into one single reward is also highly prompt-specific. We propose Reinforcement Learning with Adversarial Critic (RLAC), a post-training approach that addresses these challenges via dynamic rubric verification. Our approach employs a large language model (LLM) as a critic that dynamically identifies only the most likely failure modes (e.g., a factual error or unhandled edge case), which are then verified by an external validator to optimize both generator and critic jointly. By training both the generator and the critic, this game enhances the critic's error detection and the generator's output quality while reducing required verifications. Our experiments demonstrate that RLAC improves factual accuracy in text generation and correctness in code generation, while also outperforming exhaustive verification and reward model methods. We show that dynamic critics are more effective than fixed critics, showcasing the potential of RLAC for scaling RL post-training to free-form generation tasks.

## 1 Introduction

Post-training methods for large language models (LLMs) have progressed dramatically over the past few years, from largely manual supervised fine-tuning (SFT) techniques that rely on a combination of manual data curation (Radford et al., 2018; Brown et al., 2020; Shengyu et al., 2023) to reinforcement learning (RL) methods that perform general preference-based optimization (Christiano et al., 2017; Ouyang et al., 2022) or optimize task-specific notions of correctness (Ziegler et al., 2019; Stiennon et al., 2020). Despite these remarkable results, RL post-training is limited to tasks with clear-cut success criteria (i.e., correctness of an answer or preference of a human user), and it remains unclear how to post-train LLMs with RL on tasks that require producing open-ended or free-form outputs that are hard to verify perfectly.

Perhaps the biggest challenge in building RL post-training methods for free-form generation tasks is the lack of a solid reward function: outputs are typically expected to satisfy several task-specific rubrics. In principle, a task designer could construct a reward by combining these rubrics, but both enumerating and verifying them pose major scalability challenges (Min et al., 2023).

For instance, complex code generation requires testing countless edge cases (e.g., empty inputs or specific numbers). Even if such criteria could be enumerated, knowing how to combine them remains difficult (e.g., should correctly handling even numbers outweigh handling primes?). While RLHF-trained reward models or LLM-as-judge approaches (Christiano et al., 2017; Zheng et al., 2023) outsource the job of merging rubrics to a learned or prompted reward model, this often leads to reward hacking (Ziegler et al., 2019; Gao et al., 2023; Skalse et al., 2022; Eisenstein et al., 2023), since the best combination is highly dependent on the prompt and the model being optimized. How can we then train LLMs on free-form generation tasks with multiple (even uncountable) rubrics?

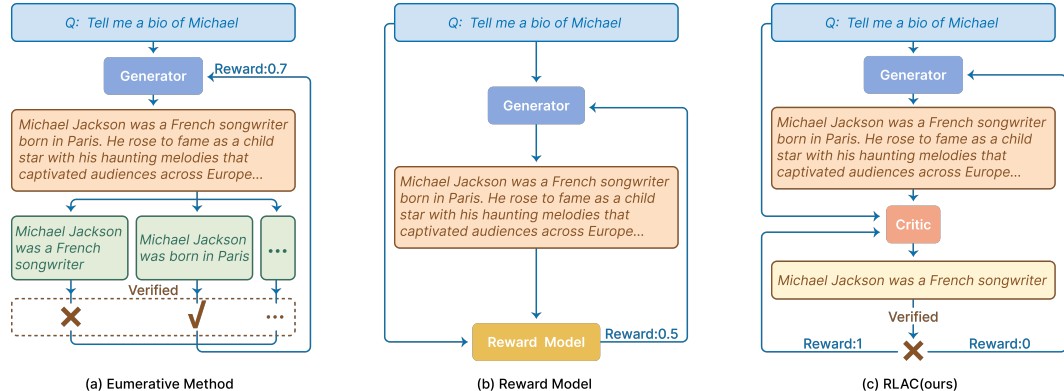

Figure 1: Comparison of three post-training paradigms on a biography example ("Michael Jackson"). **(a) Enumerative verification** explicitly extracts and checks every atomic fact before aggregating a scalar reward, which is accurate but expensive. **(b) Reward-model methods** skip verification and directly predict a scalar reward from a learned judge, which is efficient but prone to reward hacking. In contrast, **(c) RLAC** trains a learned critic to propose one likely-wrong fact (*rubric*) and verifies it via an external validator. If the fact indeed fails, the critic receives reward 1 and the generator 0; otherwise the generator receives 1 and the critic 0. This dynamic, adversarial feedback yields prompt-specific, verifiable, and scalable supervision for free-form generation tasks.

We introduce Reinforcement Learning with Adversarial Critic (RLAC), which formulates the problem as an adversarial game between a generator and a *critic*. The critic is a learned model that proposes a rubric (e.g., one test case) where the generator's output is likely to fail, and an external validator verifies this. Both models are trained jointly: the critic is rewarded when it correctly pinpoints a rubric that the generator fails (verified by an external validator), while the generator is rewarded when the critic is unable to do so. This formulation eliminates the need to enumerate or verify all rubrics, significantly improving training scalability. At the same time, it ensures that rewards are based on rubrics that are prompt-specific, adversarially chosen, and always on-policy. Figure 1 illustrates how RLAC achieves verification efficiency while maintaining accuracy through adversarial critic-generator dynamics on a biography generation example.

We evaluate Reinforcement Learning with Adversarial Critic on factual text generation and code generation, representing enumerable and non-enumerable verification scenarios, respectively. On 8-sentence biography generation with Qwen3-8B, Reinforcement Learning with Adversarial Critic achieves a FactScore of 0.889, surpassing FactTune-FS's (Tian et al., 2024) 0.867, while reducing verification calls by $5.7\times$. This efficiency gain scales with task complexity, from **4.4**$\times$ for 4-sentence to **5.7**$\times$ for 8-sentence generation. In code generation, despite using only $9\%$ of the training data, Reinforcement Learning with Adversarial Critic achieves the highest average scores on both base models: **53.2** on Qwen2.5-Coder-7B-Base and **56.6** on Qwen2.5-Coder-7B-Instruct, outperforming prior methods AceCoder-RM and AceCoder-Rule (Zeng et al., 2025).

Our primary contribution is Reinforcement Learning with Adversarial Critic (RLAC), a novel post-training paradigm that frames free-form LLM optimization as an adversarial game between a generator and a learned critic, with an external validator providing ground-truth feedback. This design avoids exhaustive rubric enumeration and mitigates reward hacking by producing task-specific and on-policy training signals. In experiments, Reinforcement Learning with Adversarial Critic consistently improves factual accuracy while reducing verification costs and surpasses prior methods on code generation, demonstrating scalable gains across both enumerable and non-enumerable verification tasks.

## 2 PRELIMINARIES

Our goal is to train an LLM generator that produces a free-form output satisfying requirements of the underlying task, without manually enumerating every rubric for evaluation and grading. In this section, we formalize this problem, introduce notation, and briefly discuss related concepts of reward models (Christiano et al., 2017; Ziegler et al., 2019; Rafailov et al., 2023) and enumerative verification (Min et al., 2023; Trivedi et al., 2024; Saha et al., 2025; Wang et al., 2024b; Xie et al., 2025). We illustrate these in Figure 1. We then present our approach in the next section.

**Problem setup.** We consider free-form generation tasks where outputs must satisfy many task-specific requirements, which we refer to as rubrics. For instance, a biography generation task may require that each factual claim is correct, while a code generation task may require the program to handle all edge cases correctly. A math proof may require proving a certain set of intermediate results. To captures these ideas abstractly, let $\mathcal{S}$ be a distribution over prompts or instructions that may be presented to an LLM. Given $s \in \mathcal{S}$, a generator LLM $\pi^g(a \mid s)$ is tasked with producing a textual output $a \in \mathcal{A}$. We choose to use standard notation typically used in RL ($\mathcal{S}$ denoting the state space and $\mathcal{A}$ denoting the action space) as we later present an RL training objective. Each instruction $s$ is inherently associated with a set of rubrics (denoted as $\mathcal{C}(s)$), where each rubric $c \in \mathcal{C}(s)$ represents a verifiable property the output should satisfy, such as *"the claim about Newton's birth year is correct"* for biography generation or *"the code handles null inputs"* for code generation.

We assume access to a binary verification or reward function $R(s, a, c)$ that returns 1 if a generated output $a \sim \pi^g(\cdot|s)$ satisfies the rubric $c$ on instruction $s$, and returns 0 otherwise. An output $a$ is considered correct only when *all* rubrics $\mathcal{C}(s)$ associated with instruction $s$ are satisfied. Our goal is to train $\pi^g(\cdot|s)$ to maximize the probability of producing fully correct outputs:

$$\pi_g^* := \arg\max_{\pi} \mathbb{E}_{s \sim \mathcal{S}} \left[ \mathbb{E}_{a \sim \pi(\cdot|s)} \Big[ \prod_{c \in \mathcal{C}(s)} R(s, a, c) \Big] \right]. \tag{1}$$

In constrained domains with a single, well-defined rubric (e.g., matching a reference final answer in math reasoning), we can solve this opimization problem via standard RL algorithms like PPO (Schulman et al., 2017) or GRPO (Shao et al., 2024). Note that these RL algorithms require evaluating these rubrics on every sample drawn from the policy. However, such cases are rare in open-ended tasks with diverse rubrics. In these settings, $\mathcal{C}(s)$ can be extremely large or even unbounded, making Eq. 1 computationally intractable since every output must be checked against every rubric.

**Reward models and enumerative verification.** Most approaches to optimizing free-form generation tackle the challenge of diverse rubrics through two paradigms. RLHF (Christiano et al., 2017) trains a single proxy reward model from offline human preference data. While efficient, this optimization is hard because the learned proxy is only as good as its coverage of the preference dataset. When the generator explores beyond this support, the proxy can misalign (Gao et al., 2023), often necessitating additional constraints like KL regularization to avoid collapse. These constraints stabilize training but limit exploration, making it difficult to scale to highly free-form generation tasks (Dong et al., 2024).

Another approach is to enumerate the evaluation criteria and optimize their aggregate, either through prompting (Min et al., 2023; Saha et al., 2025) or via preferences implicitly elicited from humans (Wang et al., 2024b; Mahan et al., 2024). While more faithful to the underlying rubrics (Trivedi et al., 2024), this strategy is fundamentally limited: it assumes the evaluation set $\mathcal{C}(s)$ can be exhaustively listed, which is unrealistic for complex tasks (e.g., all test cases for a nontrivial program). Even when such enumeration is feasible, iterating over the entire set is computationally prohibitive, turning optimization into an intractable verification bottleneck. In practice, the enumeration procedure itself can introduce bias in the aggregated reward (Appendix A).

## 3 Reinforcement Learning with Adversarial Critic

We now introduce our RL post-training approach, called Reinforcement Learning with Adversarial Critic (RLAC) for training LLM generators on free-form tasks. Our goal is to provide rewards while avoiding the scalability limits of enumerative verification and the misalignment of static reward models. The core idea is to recast verification of a generator response as a *dynamic* process guided by a learned critic. Concretely, we frame training as a two-player game: given an output from the generator, the critic proposes a rubric the output is likely to violate, while the generator aims to satisfy all such rubrics. An external validator then adjudicates whether the output meets the proposed rubric, and this supervision updates both generator and critic. In this way, verification becomes adaptive and adversarial, tailored to the generator's current weaknesses. We now formally derive this approach.

### 3.1 Problem Reformulation

To derive our approach formally, our starting point is the objective of Equation 1, which requires a generation to satisfy all rubrics in the set $\mathcal{C}(s)$: Since $R(s, a, c)$ is an indicator function for each $c$,

we can rewrite the requirement that all rubrics are satisfied as a minimum over all rubrics as follows:

$$\mathbb{1}\{R(s, a, c) = 1, \forall c \in \mathcal{C}(s)\} = \min_{c \in \mathcal{C}(s)} \mathbb{1}\{R(s, a, c) = 1\}. \tag{2}$$

Intuitively, the minimum selects the worst-case criterion, i.e., the first failure mode encountered by the current model $\pi$. Substituting Equation 2 into Equation 1 gives:

$$\pi_g^* = \arg\max_{\pi} \ \mathbb{E}_{s \sim \mathcal{S}} \left[ \mathbb{E}_{a \sim \pi(\cdot|s)} \left[ \min_{c \in \mathcal{C}(s)} R(s, a, c) \right] \right]. \tag{3}$$

However, this reformulation by itself does not make the optimization problem simpler: searching over $\mathcal{C}(s)$ is infeasible when $\mathcal{C}(s)$ is large or infinite (e.g., all possible test cases). To address this, we introduce a critic $\pi^c$, modeled as a stochastic policy that takes an instruction–generation pair (s,a) as input and outputs a rubric c $\in \mathcal{C}(s)$ in natural language, representing a verifiable property that may fail. An external validator then checks the proposed rubric. Then we can rewrite Equation 3 into the equivalent min-max form:

$$\pi^g = \arg\max_{\pi} \ \min_{\pi^c} \ \mathbb{E}_{s \sim \mathcal{S}} \left[ \mathbb{E}_{a \sim \pi(\cdot|s)} \mathbb{E}_{c \sim \pi^c(\cdot|s,a)} \left[ R(s, a, c) \right] \right]. \tag{4}$$

It can be shown that the solution $\pi^g$ from Equation 4 is the same as that from Eq. (1), but now we bypass the need to enumerate all criteria over $\mathcal{C}(s)$ (Madry et al., 2018).

Pretty much like other mini-max optimization problems, we can solve the above optimization problem by iteratively updating $\pi^g$ and $\pi^c$ against each other. The optimization goal is to achieve a robust generator $\pi^g$ that does well even according to the most adversarial critic, upon convergence. More details with respect to the practical optimization algorithm will be provided in Section 3.2.

## 3.2 PRACTICAL INSTANTIATION OF RLAC

We now instantiate the two-player adversarial game from the previous section into a practical approach that we can use to train LLMs. As shown in Figure 1, we parameterize three task-agnostic components that interact with each other during RL training. Each component is instantiated differently based on the domain (as detailed in Section 4).

**Generator.** The generator $\pi^g$, is an LLM that is fine-tuned to produce an output $a \in \mathcal{A}$ for an instruction $s \in \mathcal{S}$. RLAC samples multiple response generations from $\pi^g$ for each instruction $s$. We train $\pi^g$ to maximize the probability of producing outputs that satisfy all task-specific rubrics. The prompt for the generator is included in the Appendix B.1.

**Critic.** Our critic $\pi^c$ is a pre-trained LLM that RLAC fine-tunes. Specifically, for each instruction $s$ and a query generation output $a$, the critic is prompted to generate a natural language output representing a rubric $c$ through auto-regressive decoding. The rubric $c$ along with the instruction $s$ and the generation $a$ are then sent to the external validator to obtain a reward signal $R(s, a, c) \in \{0, 1\}$. The prompt for the adversarial critic is included in the Appendix B.2.

**Validator.** The validator is an external tool or process that can verify whether a generated response satisfies a rubric provided as input to it. Neither the generator nor the critic ever observes the reference solution or any intermediate traces. The validator can be implemented in various ways depending on the domain, such as rule-based checkers or a software tool that evaluates a proposed code on a proposed test-case. Implementation details for specific tasks are discussed in the Appendix C.

**Updating the generator and critic.** At each training step, we sample instructions $s \in \mathcal{S}$ and have the generator $\pi^g$ produce $K$ candidate outputs $a_1, \ldots, a_K$. For each $(s, a_i)$, the adversarial critic $\pi^c$ proposes a criterion $c_i$, which is then checked by the validator to yield a binary reward $r_i \in \{0, 1\}$. This online feedback provides signals for both the generator and the critic. Outputs with $r_i = 1$ are treated as positives ($a^+$) and those with $r_i = 0$ as negatives ($a^-$), and the generator is updated using the DPO objective (Rafailov et al., 2023) with respect to the reference generator $\pi_{\text{ref}}^g$:

$$\mathcal{L}(\pi^g; \pi_{\text{ref}}^g) = -\mathbb{E}_s \mathbb{E}_{(a^+, a^-)} \left[ \log \sigma \left( \beta \log \frac{\pi^g(a^+|s)}{\pi_{\text{ref}}^g(a^+|s)} - \beta \log \frac{\pi^g(a^-|s)}{\pi_{\text{ref}}^g(a^-|s)} \right) \right]. \tag{5}$$

We adopt DPO primarily for its simplicity and stability, it allows direct policy optimization from binary preference signals without requiring explicit reward scaling or KL-penalty tuning. Importantly,

---

**Algorithm 1** RLAC

---

1: Initialize parameters $\pi^g, \pi^c, \pi^g_{\text{ref}}, \pi^c_{\text{ref}}$
2: **for** each iteration **do**
3:      ## Policy Evaluation for Generator $\pi^g$.
4:      **for** each instruction $s$ **do**
5:          Generate $K$ generations $a_1, ..., a_K \sim \pi^g(\cdot|s)$
6:          Sample a criterion from the adversarial critic for each generation $c_i \sim \pi^c(\cdot|s, a_i)$.
7:          Construct a generator dataset $\mathcal{D}^g_s = \{(s, a_i, R(s, a_i, c_i))\}^K_{i=1}$
8:      ## Policy Evaluation for Critic $\pi^c$.                          ▷ Optional
9:      **for** each instruction $s$, output $a$ **do**
10:         Generate $N$ criteria $c_1, ..., c_N \sim \pi^c(\cdot|s, a)$
11:         Construct a critic dataset $\mathcal{D}^c_{(s,a)} = \{(s, a, R(s, a, c_j))\}^N_{j=1}$
12:      ## Policy Improvement for Generator $\pi^g$.
13:      $\pi^g_{\text{new}} \leftarrow \pi^g$
14:      **for** each update step **do**
15:         $\pi^g_{\text{new}} \leftarrow \pi^g_{\text{new}} - \nabla\mathcal{L}(\pi^g_{\text{new}}, \pi^g_{\text{ref}})$                   ▷ Equation 5
16:      $\pi^g_{\text{ref}} \leftarrow \pi^g$
17:      ## Policy Improvement for Critic $\pi^c$.                     ▷ Optional
18:      $\pi^c_{\text{new}} \leftarrow \pi^c$
19:      **for** each update step **do**
20:         $\pi^c_{\text{new}} \leftarrow \pi^c_{\text{new}} - \nabla\mathcal{L}(\pi^c_{\text{new}}, \pi^c_{\text{ref}})$                 ▷ Equation 6
21:      $\pi^c_{\text{ref}} \leftarrow \pi^c$

---

RLAC is agnostic to the choice of policy optimization algorithm: Any online or offline RL objective (e.g. PPO (Schulman et al., 2017), GRPO (Shao et al., 2024)) can be substituted here without affecting the overall framework, since the critic–validator loop provides compatible supervision in all cases.

Similarly, for each $(s, a)$ pair, we sample $N$ criteria from $\pi^c$. Criteria rejected by the validator (invalid or satisfied by the generator) are treated as negatives ($c^-$), while valid, unsatisfied ones are positives ($c^+$). The critic is then updated with the same DPO objective relative to its reference policy $\pi^c_{\text{ref}}$:

$$\mathcal{L}(\pi^c; \pi^c_{\text{ref}}) = -\mathbb{E}_{s,a}\mathbb{E}_{(c^+,c^-)}\left[\log\sigma\left(\beta\log\frac{\pi^c(c^+|s,a)}{\pi^c_{\text{ref}}(c^+|s,a)} - \beta\log\frac{\pi^c(c^-|s,a)}{\pi^c_{\text{ref}}(c^-|s,a)}\right)\right]. \quad (6)$$

In this way, evaluation and improvement are unified: the critic adaptively identifies failure modes, the validator provides ground-truth feedback, and both generator and critic are jointly updated to improve over time.

**Algorithm summary.** Algorithm 1 summarizes the practical implementation of RLAC. At a high level, the algorithm follows a standard online RL loop that alternates between policy evaluation and improvement. In each evaluation step, we sample generations from the current generator $\pi^g$, have the critic propose a criterion $c$, and obtain verification to assign rewards. These rewards are then used to update the generator with the DPO objective (Equation 5). Optionally, we also collect evaluation data for the critic by sampling multiple criteria per instruction–generation pair. The critic is then updated with its own DPO objective (Equation 6), allowing it to adaptively identify weaknesses in the generator and provide more effective learning signals.

## 4 EXPERIMENTS

We now evaluate our approach on two free-form generation tasks: factual text generation (§4.1) and code generation (§4.2). Factual text generation presents the enumerable-but-expensive regime, where all claims can, in principle, be verified but at a cost that scales with length of the text. This tests RLAC's ability to maintain verification quality while reducing calls. Code generation, by contrast, represents the non-enumerable regime, where exhaustive verification is impossible due to infinite corner cases and intractable formal checks (Church, 1936). Here, the goal is to expose critical failures through targeted critic proposals. Together, these tasks span the spectrum from costly-but-possible to fundamentally intractable verification, highlighting the broad applicability of RLAC.

Table 1: Performance comparison on factual text generation. RLAC achieves the highest FactScore across all settings while using fewer verification calls than FactTune-FS.

| Method | 4-sentence Generation | | | | 8-sentence Generation | | | |
|---|---|---|---|---|---|---|---|---|
| | # Corr↑ | # Incorr↓ | FS↑ | Calls↓ | # Corr↑ | # Incorr↓ | FS↑ | Calls↓ |
| *Qwen3-4B* | | | | | | | | |
| Baseline | 10.07 | 6.43 | 0.610 | - | 19.62 | 12.08 | 0.619 | - |
| FactTune-FS | 10.66 | 3.48 | 0.754 | 214,911 | 20.65 | 5.99 | 0.775 | 341,657 |
| ArmoRM | **14.54** | 8.69 | 0.626 | - | 21.02 | 10.02 | 0.677 | - |
| RLAC (Ours) | 10.54 | **3.04** | **0.776** | **57,600** | 21.58 | **4.84** | **0.817** | **48,000** |
| *Qwen3-8B* | | | | | | | | |
| Baseline | 12.65 | 5.53 | 0.696 | - | 22.51 | 11.97 | 0.653 | - |
| FactTune-FS | **13.31** | 3.63 | 0.786 | 168,735 | **25.10** | 3.84 | 0.867 | 438,949 |
| ArmoRM | 12.96 | 6.86 | 0.654 | - | 23.31 | 8.92 | 0.723 | - |
| RLAC (Ours) | 13.14 | **3.37** | **0.796** | **38,400** | 24.33 | **3.03** | **0.889** | **76,800** |

## 4.1 FACTUAL TEXT GENERATION

**Evaluation data & metrics.** We follow Min et al. (2023); Tian et al. (2024) in adapting a factual text generation task in which the model should produce concise biographies for a given individual. We use 170 topics from the Wikipedia Biography Dataset (Lebret et al., 2016), split into 120 for training and 50 for testing.

We use factual precision of the output (as defined by FactScore (Min et al., 2023)) as the primary metric, and also report the counts of correct and incorrect facts. To control for length, the model is instructed to generate either four or eight sentences. Since frequent calls to the external validator are costly, we additionally track the number of validator calls.

**Base models & baselines.** We compare RLAC against two dominant paradigms of post-training: (1) enumerative verification, which relies on explicit checking of all atomic facts, and (2) reward-model optimization, which replaces external verification with a learned scalar judge. To represent these paradigms, we evaluate the following baselines and prior approaches: (1) the Qwen3-4B and Qwen3-8B base models as starting generators for our study; (2) FactTune-FS (Tian et al., 2024), a widely used method for factual text generation to represent exhaustive verification using an external validator, FactScore, for all atomic facts; and (3) ArmoRM (Wang et al., 2024a), which represents the reward model based method that produces one reward score for the generated output.

Both the generator and critic are initialized from the same backbone models (Qwen3-4B and Qwen3-8B) to ensure fairness. We use FactScore as an external validator, i.e., FactScore checks whether a critic-proposed fact appears in the biography and is correct according to Wikipedia. All methods are trained with multiple rounds of DPO updates, where the generator produces 10 outputs per prompt and the critic proposes 4 rubrics per output. These values follow common configurations in preference-based optimization (Rafailov et al., 2023; Tian et al., 2024) and provide a balanced trade-off between exploration diversity and verification cost, ensuring fair comparison across methods (see Appendix D for a detailed analysis of these hyperparameters).

**Results.** Table 1 shows that RLAC achieves the highest factuality scores across model sizes and output lengths, while using significantly fewer verification calls. For instance, on Qwen3-8B with eight-sentence generation, it reaches a FactScore of 0.889, outperforming FactTune-FS (0.867) and ArmoRM (0.723), but with only 77k verification calls compared to 439k for FactTune-FS. This efficiency gap grows with output length: FactTune-FS requires $4.4\times$ more verification calls in the four-sentence setting (169k vs. 39k) and $5.7\times$ more in the eight-sentence setting (439k vs. 77k). This shows that RLAC scales more efficiently as the generation complexity increases. Further results demonstrating robustness across multiple random seeds, as well as generalization to the MedicalQA dataset are detailed in Appendices E and G.

**RLAC's improvements throughout training.** Figure 2 shows how the generator's accuracy evolves over training, measured along three axes: training epoch, number of verification calls, and KL divergence from the base model. In Figure 2(a), RLAC shows a slight initial drop in FactScore (from 0.653 to 0.641). At this early stage, the critic has not yet learned to identify the most obvious errors, so

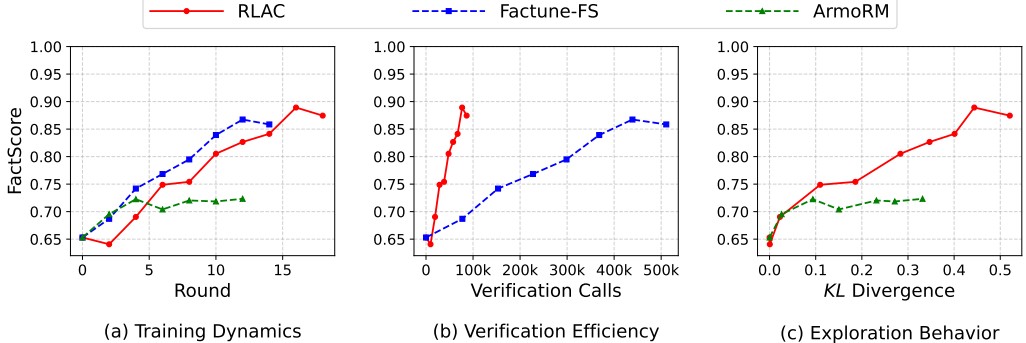

Figure 2: Comparison of training dynamics, verification efficiency, and exploration behavior for RLAC, FactTune-FS, and ArmoRM on the Qwen3-8B model with 8-sentence generation.

Table 2: Generator's test accuracy across critic types.

| Method | # Corr | # Incorr | FS |
|---|---|---|---|
| Base | 19.62 | 12.08 | 0.619 |
| Noisy Validator | 19.84 | 12.83 | 0.607 |
| Static Critic | 17.77 | **3.77** | **0.825** |
| Adversarial Critic | **21.58** | 4.84 | 0.817 |

the "mistakes" it proposes are often minor or even incorrect. As a result, the generator receives weak targeted training signals, and factuality temporarily degrades. After several rounds, the critic improves at detecting mistakes, which in turn accelerates generator learning. Once this dynamic stabilizes, the generator's factuality gradually improves, ultimately reaching 0.889, outperforming FactTune-FS (0.867). This two-phase process illustrates how RLAC evolves from weak initial supervision to highly efficient, targeted verification.

Figure 2(b) shows that RLAC achieves the same level of factuality as FactTune-FS with far fewer verification calls (e.g., 67K vs. 368K to achieve 84%). This highlights the inefficiency of FactTune-FS, which repeatedly validates already correct facts, whereas RLAC dynamically targets high-risk errors, yielding greater verification efficiency and scalability.

Figure 2(c) measures exploration by tracking the KL divergence from the base model. Such deviation can usually be caused by either (1) improvements from the base model through effective exploration, or (2) reward hacking, in which the model overfits to the reward model and drafts without real quality gains. For RLAC, KL increases alongside monotonic FactScore gains ($0.653 \rightarrow 0.889$), indicating productive exploration. In contrast, RL with a fixed offline reward model (ArmoRM) shows a rise in KL without the corresponding factuality gains, evidence of reward hacking. These dynamics (which are not backbone-specific, see Appendix F) complement Table 1: while both RLAC and FactTune-FS improve factuality, RLAC achieves comparable or higher FactScore with far fewer verification calls, whereas ArmoRM inflates output length without consistent accuracy due to its static reward.

**Ablation study.** We compare RLAC with two ablated variants to isolate the factors driving its effectiveness. In the first, we replace the external validator's outputs with random correctness labels to assess the role of validator reliability. In the second, we freeze the critic model, referred to as a *static* critic rather than training it adversarially with the generator, to evaluate the importance of adversarial joint training.

As shown in Table 2, noisy validation destabilizes training and reduces performance below the base model, highlighting the importance of reliable validation. The static critic achieves a superficially high FactScore by generating fewer facts, reducing both correct and incorrect facts, unlike the adversarially trained critic that increases correct facts while reducing errors. This indicates that the static critic inflates precision rather than genuine factual improvement. Figure 3 further illustrates these dynamics. The static critic's validator outcomes quickly rise to about 0.81, showing that the generator quickly learns to evade its fixed patterns. In contrast, the adversarial critic's outcomes grow much more slowly and reach a lower level of about 0.6 by round 16, indicating that it continues to surface genuine errors and sustain learning pressure. In general, these results highlight that both reliable external

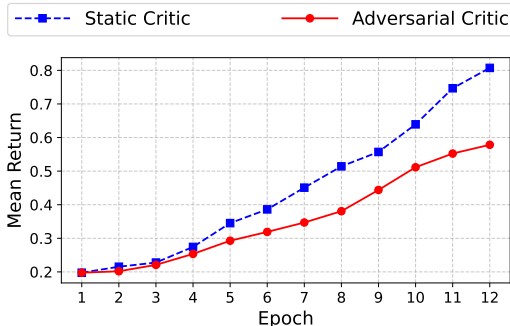

Figure 3: Average validator outcomes on suspicious facts proposed by the critic during factual biography generation. Higher values indicate that the critic more often misjudges correct facts (i.e., weaker supervision).

verification and a dynamically adapting critic are crucial: Without either, the generator fails to achieve meaningful gains in factual accuracy, validating the core design of RLAC.

## 4.2 CODE GENERATION

**Evaluation data & metrics.** We evaluate code generation performance using widely studied benchmarks: HumanEval (Base and Plus) (Chen et al., 2021; Liu et al., 2023), MBPP (Base and Plus) (Austin et al., 2021; Liu et al., 2023), BigCodeBench (Zhuo et al., 2024), and LiveCodeBench (V4) (Jain et al., 2025). We use Pass1 as a primary metric. For efficiency analysis, we also report the total number of test cases executed during RL training as the number of validator calls.

**Base models & baselines.** For training data, we use the AceCode-87K-hard subset (Zeng et al., 2025), consisting of approximately 22K problems. We compare against the following baseline and prior methods: (1) the base model Qwen2.5-Coder-7B-Base and Qwen2.5-Coder-7B-Instruct without training; (2) an enumerative RL method AceCoder-Rule, which employs RL with rule-based binary rewards from test execution; and (3) a reward model method AceCoder-RM, which uses RL with AceCodeRM-7B trained on approximately 300K preference pairs from AceCode-87K dataset.

For all methods, we follow the AceCoder experimental setup using the AceCode-87K-hard subset (Zeng et al., 2025), which contains about 22K problems and generates $k = 8$ outputs per prompt. For training efficiency, our method samples 2k problems randomly from this subset and uses $n = 2$ critic proposals per generation (see Appendix D). Note that, unlike factual verification, where each check is expensive, unit tests in code generation are cheap to execute. The fundamental bottleneck here is not the per-test cost but the non-enumerability of the test space. Therefore, test-case count is reported only for completeness; the primary evaluation metric remains Pass@k.

**Results.** Table 6 summarizes results across five widely-used code generation benchmarks. Despite training on only 2,000 problems (9% of the dataset used for AceCoder-RM and AceCoder-Rule), RLAC achieves the highest average scores: 53.2 using Qwen2.5-Coder-7B-Base and 56.6 using Qwen2.5-Coder-7B-Instruct, consistently outperforming both enumerative method (AceCoder-Rule) and static reward model method (AceCoder-RM) across the majority of benchmarks. Similarly, while AceCoder-Rule executed approximately 7.86 million test cases during training, RLAC required 192 thousand to reach higher final performance, reducing verification cost by 97.5%.

We observe from Table 6 that AceCoder-RM not only fails to improve performance but can even degrade it under noisy validation. For example, on HumanEval, performance drops from 91.5 to 89.0 despite using the competetive reward model Acecoder-RM-7B, indicating reward hacking.

This fragility arises from the reward model trained on preference pairs from the AceCoder dataset, which itself contains noisy and incomplete test cases (Zeng et al., 2025). During RL training, as the generator's outputs drift away from the RM's fixed training distribution, these noisy supervision signals are further amplified. The static RM cannot adapt, causing it to favor spurious correlations rather than true correctness, leading the generator to exploit flaws in the reward signal.

Table 3: Results for HumanEval, MBPP, BigCodeBench Complete and Instruct (BCB-C, BCB-I), and LiveCodeBench, using two different base models. RLAC achieves the highest average score across benchmarks.

| Method | HumanEval | | MBPP | | BCB-C | | BCB-I | | LCB | Average |
|---|---|---|---|---|---|---|---|---|---|---|
| | Base | Plus | Base | Plus | Full | Hard | Full | Hard | | |
| ***Base: Qwen2.5-Coder-7B-Base*** | | | | | | | | | | |
| Baseline | 83.5 | 79.3 | 80.4 | 69.3 | 45.8 | 16.2 | 40.2 | 14.2 | **28.7** | 50.8 |
| AceCoder-RM | 83.5 | 75.6 | 80.2 | 67.2 | 41.9 | 14.9 | 36.8 | 16.2 | 25.7 | 49.1 |
| AceCoder-Rule | 84.1 | 78.0 | 82.3 | 69.3 | 48.6 | 18.2 | **43.2** | **18.2** | 28.5 | 52.3 |
| RLAC (Ours) | **85.7** | **80.6** | **82.4** | **71.6** | **50.3** | **20.9** | 42.1 | 16.9 | 28.7 | **53.2** |
| ***Base: Qwen2.5-Coder-7B-Instruct*** | | | | | | | | | | |
| Baseline | 91.5 | 84.8 | 82.8 | 71.4 | 49.5 | 19.6 | 41.8 | **20.3** | 34.2 | 55.1 |
| AceCoder-RM | 89.0 | 84.1 | **86.0** | 72.8 | 50.4 | 18.9 | 42.0 | 19.6 | 35.0 | 55.3 |
| AceCoder-Rule | 90.9 | 84.8 | 84.1 | 71.7 | 50.9 | 23.0 | **43.3** | 19.6 | 34.9 | 55.9 |
| RLAC (Ours) | **93.3** | **86.0** | 83.9 | **73.0** | **52.2** | **24.3** | 42.3 | 19.6 | **35.2** | **56.6** |

RLAC also suffers from the noisy dataset since we use a simulated 8 as validator mentioned in settings. Although the critic is also affected by noise, its continuous adaptation allows it to stay aligned with the generator's changing behavior, preserving meaningful supervision. As a result, RLAC consistently improves performance across all benchmarks, even in noisy and imperfect validation environments, showing robustness to noisy validation.

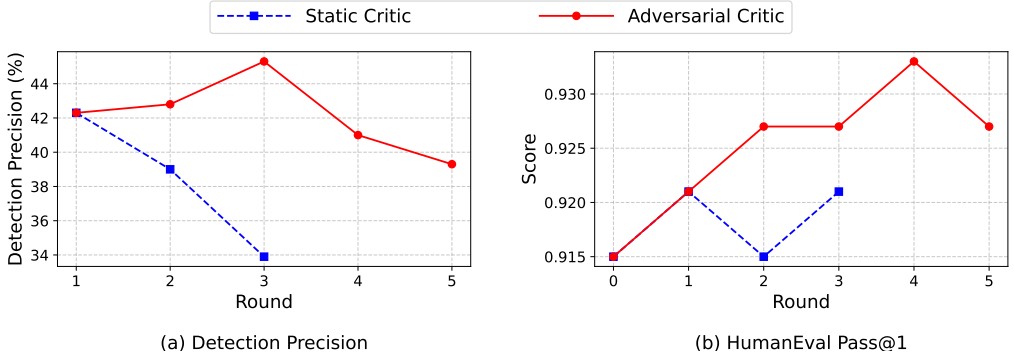

(a) Detection Precision          (b) HumanEval Pass@1

Figure 4: **Ablations on the static critic vs. adversarial critic.** Static critic's detection accuracy degrades from 42.3% to 33.9% as the generator exploits its patterns, yielding minimal performance gains (+0.6%) compared to the adversarial critic's continued improvement (+1.8%).

**Ablation study.** We compare RLAC with a variant that replaces the adversarially trained critic with a static critic to evaluate the necessity of dynamic adaptation. As shown in Figure 4, the static critic's detection rate, defined by the fraction of test cases generated that correctly expose real errors, drops dramatically from 42.3% to 33.9% over three rounds, as the generator gradually learns to exploit its fixed detection patterns. In contrast, the adversarial critic maintains a stable detection rate greater than 39% by continuously adapting to the evolving behavior of the generator.

This degradation directly impacts performance: with the static critic, the generator plateaus at 92.1% Pass@1, while RLAC reaches 93.3%. Further analysis shows that static critic's test cases critic's output degenerated to minor variations of earlier ones, allowing the generator to avoid detection by simplifying or reducing outputs rather than truly fixing bugs. These results highlight that dynamic adaptation is essential for preventing reward hacking and driving real code correctness improvements.

## 5 RELATED WORKS

**Reward models.** One possibility for evaluating free-form and open-ended generations is to encode all criteria into a single scalar through a learnt reward model. This is usually achieved through learning from an offline dataset of human preferences (Christiano et al., 2017; Ziegler et al., 2019;

Yi et al., 2019; Böhm et al., 2019; Rafailov et al., 2023) or absolute ratings (Cui et al., 2024; Wang et al., 2024c). Multi-objective reward models (Wang et al., 2024a; Dong et al., 2024; Ji et al., 2023) expose several fixed dimensions (e.g., truthfulness, honesty), improving interpretability but still relying on static, globally defined criteria. Our approach differs conceptually: instead of collapsing all rubrics into a single scalar or a fixed multi-objective vector, we learn a critic that dynamically proposes a verifiable rubric for each instance and grounds its supervision through an external validator. This yields a reward signal that is still scalar for RL optimization, but derived from an objectively checkable criterion rather than a static, unverified proxy, offering better alignment and reliability in open-ended tasks.

**Enumerative verifications for free-form generations.** To obtain a comprehensive and reliable evaluation of free-form generations, the standard practice is to enumerate a set of fine-grained criteria (Zhuge et al., 2024; Min et al., 2023; Saad-Falcon et al., 2024; Chang et al., 2024; Xie et al., 2025). While they can be automatically deposed by LLMs for easier domains (Min et al., 2023; Jing et al., 2024), extensive manual annotations are typically required for more complex domains such as travel planning (Xie et al., 2024), codebase generation (Zhao et al., 2025), and research reproduction (Starace et al., 2025). Dedicated computation and actions such as information retrieval (Min et al., 2023) and code execution (Zhuge et al., 2024; Starace et al., 2025) require manual rubric design or domain-specific validators (e.g., retrieval and code execution). Because all rubrics must be checked for each output, verification cost scales roughly linearly with the number of possible rubrics and may still miss unlisted error types. In contrast, RLAC replaces exhaustive enumeration with a learned critic that dynamically selects the most informative, verifiable failure mode for each instance. By verifying only this targeted rubric via an external validator, the method retains rubric-level faithfulness while substantially reducing evaluation cost and exposing diverse, on-policy errors that static checklists often overlook. We provide additional discussion of outcome-reward RL for domains with verifiable answers in Appendix J.

**LLM-as-a-Judge.** Because of the common-sense and reasoning capabilities of pre-trained LLMs, they can directly be prompted to serve as a judge to evaluate free-form generations (Zheng et al., 2023; Yuan et al., 2025; Zhu et al., 2025). Their capabilities in evaluations can be further improved through explicit fine-tuning (Wang et al., 2024b; Yuan et al., 2025). They can also be more interpretable and robust by introducing a long Chain-of-Thought (CoT) reasoning to explicitly verify fine-grained criteria (Saha et al., 2025; Wang et al., 2024b; Trivedi et al., 2024). Beyond rubric-only judging, *generative verifiers* treat verification itself as next-token generation: they first produce verification rationales or counterevidence, and then score or select candidates (Zhang et al., 2025; Singhi et al., 2025; Setlur et al., 2025). These approaches, however, use the judge or verifier only as a static evaluator. They produce fixed judgments or explanations but do not learn adaptively from the generator's evolving behaviors. In contrast, RLAC treats the verifier as a learned critic policy within an adversarial training loop: the critic dynamically proposes which rubric to verify for each instance, receives direct feedback from an external validator, and updates jointly with the generator. This design transforms LLM-as-a-judge from a static scoring module into an active, on-policy agent that allocates verification effort where it is most informative.

## 6    CONCLUSION

We presented Reinforcement Learning with Adversarial Critic (RLAC), a new post-training approach for open-ended tasks requiring diverse, task-specific rubrics, where exhaustive enumeration is infeasible and optimal reward design is unknown. RLAC formulates training as an adversarial min-max game between a generator and a *critic*, a model that dynamically identifies the worst-case rubric for each output and verifies it externally. By jointly training both models, our approach bypasses the need for exhaustive verification or manual reward design while providing adaptive learning signals that prevent reward hacking. On the factual text generation task and code generation task, RLAC outperforms competitive baselines with significantly lower verification cost. Ablation studies further confirm the critical role of components such as adversarial critic training.

While we evaluate RLAC on two domains, we expect it to generalize broadly to other open-ended generation tasks where multiple evaluation criteria make exhaustive or rubric-by-rubric verification infeasible, such as story or scientific text generation. By adaptively selecting the most critical rubric at each step, RLAC makes RL training practical for complex generation tasks that were previously intractable due to the combinatorial explosion of rubrics or the lack of universal reward functions.

## ACKNOWLEDGMENTS

We thank members of the UC Berkeley RAIL lab for their support and feedback on the project. We thank Qi Wang for helpful discussions and for polishing the draft of our paper. This work was done when Mian Wu was visiting UC Berkeley.

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

# Appendices

## A    Limitations of Exhaustive Fact Decomposition and Per-Fact Verification

In factual text generation tasks, the rubric for the text can be verified exhaustively. FactScore (Min et al., 2023) and our exhaustive-verification baseline FactTune-FS (Tian et al., 2024) directly use an LLM to decompose the text into atomic facts, then verify each atomic fact independently using retrieved evidence (e.g., top relevant Wikipedia passages), and finally aggregate these per-fact judgments into a single factuality score.

However, this approach has drawbacks in terms of cost and accuracy. In this pipeline, the dominant verification cost scales linearly with the number of extracted atomic facts, i.e., the number of per-fact verification calls. In our biography settings, a 4-sentence output contains about 17 atomic facts on average, whereas an 8-sentence output contains about 30, so the cost grows quickly with output length.

Beyond cost, exhaustive fact extraction does not necessarily yield a cleaner reward signal. In practice, atomic fact decomposition can introduce overlapping or near-duplicate claims, each of which is verified independently and contributes separately to the aggregated score, thereby overweighting a single underlying statement and amplifying noise. For example, consider the sentence:

> "Aaron Kwok is a renowned Hong Kong singer and actor known for his versatile talent in both music and film."

A plausible decomposition into atomic facts may include:

- ($f_1$) Aaron Kwok is a singer.
- ($f_2$) Aaron Kwok is an actor.
- ($f_3$) Aaron Kwok is a Hong Kong singer.
- ($f_4$) Aaron Kwok is a Hong Kong actor.
- ($f_5$) Aaron Kwok is a renowned Hong Kong singer.
- ($f_6$) Aaron Kwok is a renowned Hong Kong singer and actor.
- ($f_7$) Aaron Kwok has versatile talent in both music and film.

Several items above are overlapping refinements of essentially the same claim (e.g., ($f_1$)–($f_6$)). Nevertheless, in an exhaustive pipeline they would be verified independently and would each influence the final aggregated score, potentially making the resulting reward noisy and sensitive to decomposition artifacts.

## B    Prompts

This section lists the exact prompts used for the **generator model** and **critic model** during data creation and training. They correspond to the input format described in Section 3.2 of the main paper.

### B.1    Generator Prompt

**Factual Text Generation**

System message: You are an AI assistant that provides accurate and concise biographies of individuals. Each biography should be exactly four sentences long, highlighting key aspects of the person's life, achievements, and significance.

User message: Write a biography of topic.

**Code Generation**

```
The generator input exactly matches the problem statement provided in
TIGER-Lab/AceCode-87K-hard without modification.

User message:
{problem_statement_from_AceCode-87K-hard}
```

## B.2 CRITIC PROMPT

**Factual Text Generation**

```
System message:
You are a factual checker. Based on your existing knowledge,
identify exactly one sentence that contains the most clearly
verifiable factual error in the paragraph.
Return your answer in **exactly three lines**:
reason:  < briefly explaining what is wrong >
sentence: N       N is the number of the most incorrect sentence
(positive integer)
error_fact: F     a brief clause (no more than 8 words) capturing the
wrong claim from that sentence

User message:
Here is an example to show the task.
Find the sentence that contains the most clearly verifiable factual error
in the paragraph about Albert Einstein.

Example paragraph:
[1] Albert Einstein was awarded the Nobel Prize in Physics in 1921 for
    his discovery of the photoelectric effect.
[2] He was born in New York City, United States, and later moved to
    Europe where he continued his studies.
[3] Einstein developed the theory of relativity, revolutionizing our
    understanding of space, time, and gravity.
[4] His famous equation describes the equivalence of mass and energy.

Expected answer:
reason: Einstein was actually born in Ulm, Germany, not New York City.
sentence: 2
error_fact: Albert Einstein was born in New York City.

Now apply the same procedure to the paragraph below about {topic}.

Paragraph:
{numbered_paragraph}

Answer:
```

**Code Generation**

```
System message:
You are a code critic. Analyze code for bugs and generate failing test
cases.
Strictly follow the format with <think> and <testcase> tags.

User message:
Analyze the given problem and the generated code to find a test case that
would cause the code to fail.

Problem: {question}

Generated code:
```python
{code}
```

```
```
First, think through potential bugs and edge cases in <think> </think>
tags.
Then output exactly ONE failing test case inside <testcase> tags using
this format:

Option A (CALL format)
<testcase> CALL: func_name(arg1, arg2, kw=val) </testcase>
Option B (STDIN format)
<testcase> STDIN: <raw input here> </testcase>

Do NOT include expected outputs or explanations.
{optional_examples_block}
```

## C  VALIDATOR IMPLEMENTATION DETAILS

This section provides the detailed design of the validator used in our training pipeline, corresponding to Section 3.2 of the main paper.

### C.1  FACTUAL TEXT GENERATION

We follow a strict validation process to ensure both authenticity and factual accuracy. In the first stage, the critic outputs both a suspected erroneous fact and the sentence number containing it. To prevent exploitation through information injection, we use textual entailment checking to verify that the proposed fact genuinely appears in the specified sentence. In the second stage, for proposals passing authenticity checks, we reuse FactScore's atomic fact verification component, which queries Wikipedia knowledge base to provide binary verification of individual factual claims, returning true or false based on external verification.

### C.2  CODE GENERATION

Since the AceCoder dataset lacks reference solutions to prevent data contamination, we construct reliable verification anchors by using Qwen2.5-Coder-7B-Instruct to generate solutions. To further rule out same-family leakage, we also generate reference solutions using GPT-4o and observe similar improvements (Appendix H). We filter these solutions using original test cases, retaining only those that pass, yielding a set of highly reliable anchors (99.7% pass rate) that serve as simulated ground truth for validating critic-proposed test cases. Given a critic-generated test case, we first execute it on the reference solution to check that the test case is valid (i.e., it executes and yields a well-defined output) and to obtain the expected output, and then execute it on the model-generated code to obtain the actual output. We define the validation reward as $R(s, a, c) = 1$ if the outputs match and 0 otherwise, treating execution failures as detected errors. The AceCoder dataset contains noise in GPT-4o generated test cases, which introduces some bias in our reference-based validator but reflects realistic imperfections in verification tools.

## D  ANALYSIS OF $K$ AND $N$

This section clarifies the roles of the hyperparameters $K$ and $N$ in RLAC and explains why their values differ between the factual and code generation experiments.

In RLAC, the parameter $K$ controls how many candidate outputs are sampled for each prompt. A larger $K$ increases candidate diversity and raises the probability that, for the same instruction, at least one candidate passes all critic checks while another fails at least one. This is essential for constructing non-degenerate preference pairs for DPO, since each pair requires both a "chosen" and a "rejected" candidate.

The hyperparameter $N$ specifies how many criteria or testcases the critic proposes for each candidate. A larger $N$ expands the critic's search space and enables it to discover more potential failure modes. However, this comes with two main drawbacks: fewer candidates pass all checks (which

reduces preference pairs available for training generator), and verification costs increase substantially. Additionally, an excessively large $N$ may introduce redundant checks without meaningful benefits.

For factual text generation, we set $K = 10$, following the configuration used in FactTune-FS, our main baseline. For code generation, we use $K = 8$, consistent with the AceCoder setup. Thus, the difference in $K$ is not arbitrary; it adheres to the standard experimental settings established in prior work.

The choice of $N$ is primarily determined by the structure of the underlying verification space. In the factual generation task, the rubric space used by FactScore is large and semantically diverse, and the biography dataset itself is relatively small. Therefore, a moderately larger $N$ helps the critic explore a broader range of failure modes. In contrast, the code generation dataset contains over 2,000 problems, and the unit tests used as criteria are typically deterministic. As a result, a smaller $N$ is already sufficient to generate a large number of informative preference pairs.

## E  QUANTIFICATION OF UNCERTAINTY

Table 4: Factual text generation on 8-sentence biographies with the Qwen3-4B backbone, with results shown as mean ± standard deviation across three runs.

| Method | # Corr↑ | # Incorr↓ | FS↑ | Calls↓ |
|---|---|---|---|---|
| | \multicolumn{4}{c}{8-sentence Generation} | | | |
| **Qwen3-4B** | | | | |
| Baseline | 19.03 ± 0.41 | 12.05 ± 0.19 | 0.616 ± 0.006 | - |
| FactTune-FS | 21.67 ± 0.79 | 5.66 ± 0.48 | 0.793 ± 0.017 | 402,781 ± 52,264 |
| ArmoRM | **22.51 ± 1.00** | 9.39 ± 0.58 | 0.705 ± 0.013 | - |
| RLAC (Ours) | 21.45 ± 0.31 | **4.37 ± 0.48** | **0.831 ± 0.016** | **70,667 ± 20,072** |

We report variability across random seeds for the factual text generation experiment. For each method, we run training with three different random seeds (affecting data shuffling and sampling of generator candidates) and report the mean and standard deviation of the metrics. Table 4 summarizes the results for the 8-sentence biography setting with Qwen3-4B as the base model. We observe that RLAC consistently improves the number of correct biographies and the overall FactScore compared to the baselines, while using substantially fewer verification calls than FactTune-FS. The standard deviations are relatively small, indicating that the performance gains are robust across independent runs.

## F  BACKBONE ROBUSTNESS

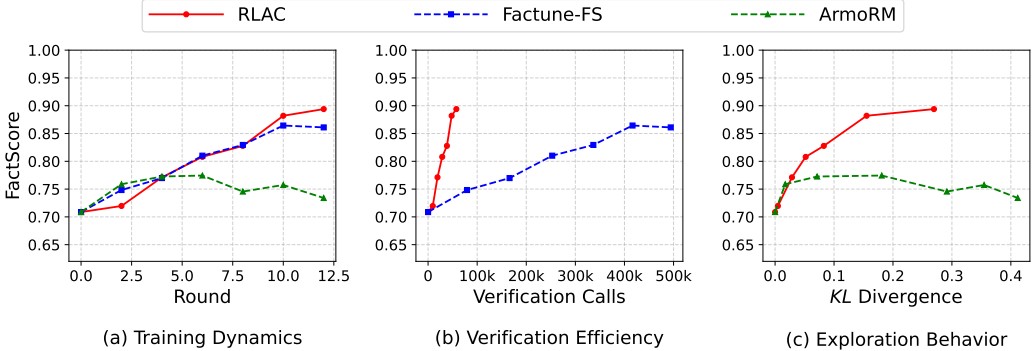

(a) Training Dynamics  (b) Verification Efficiency  (c) Exploration Behavior

Figure 5: Comparison of training dynamics, verification efficiency, and exploration behavior for RLAC, FactTune-FS, and ArmoRM on the LLaMA-3.1-8B-Instruct model with 8-sentence generation.

To examine whether the effectiveness of RLAC depends on a particular model family, we conducted additional experiments using an alternative generator/critic backbone, LLaMA-3.1-8B-Instruct. The

training setup strictly follows the configuration used in the Qwen experiments. Figure 5 shows the resulting training dynamics. We observe the same qualitative behavior as in the Qwen-based experiments: RLAC consistently improves factual correctness during training while requiring fewer verification calls than Factune-FS, consistent with the trend observed in the main paper; moreover, in the high-KL regime, RLAC achieves substantially greater gains in factual accuracy compared to the ArmoRM method.

# G   MEDICALQA

To demonstrate that RLAC generalizes beyond biographical data, we evaluate our method on a MedicalQA dataset. Factual generation in the medical domain presents a fundamentally different challenge, featuring specialized terminology and distinct semantic structures compared to Wikipedia biographies.

Since the authors of FactTune-FS did not release their MedicalQA dataset, we constructed our own by randomly sampling 170 common medical conditions from Wikipedia topics, split into 120 for training and 50 for testing. We maintain the same 8-sentence generation setting using the Qwen3-4B backbone.

Table 5: Factual text generation on 8-sentence Medical Question with the Qwen3-4B backbone.

| Method | 8-sentence Generation | | | |
|---|---|---|---|---|
| | # Corr↑ | # Incorr↓ | FS↑ | Calls↓ |
| *Qwen3-4B* | | | | |
| Baseline | 34.0 | 3.38 | 0.909 | - |
| FactTune-FS | 34.2 | 1.99 | 0.945 | 524329 |
| ArmoRM | 35.1 | 2.36 | 0.937 | - |
| RLAC (Ours) | **35.2** | **1.67** | **0.955** | **76800** |

As shown in Table 5, RLAC achieves the highest factual accuracy (0.955 FactScore) while requiring $6.8\times$ fewer verification calls than FactTune-FS (76,800 vs. 524,329). These results exactly mirror the efficiency and performance gains observed in the biography experiments, demonstrating the generality and robustness of our approach regardless of the underlying text domain.

# H   CROSS-FAMILY

Table 6: Results for HumanEval, MBPP, BigCodeBench Complete and Instruct (BCB-C, BCB-I), and LiveCodeBench, Using a different model (GPT-4O) to generate reference solutions, the results show that RLAC still achieves a similar level of average performance improvement across all benchmarks.

| Method | HumanEval | | MBPP | | BCB-C | | BCB-I | | LCB | Average |
|---|---|---|---|---|---|---|---|---|---|---|
| | Base | Plus | Base | Plus | Full | Hard | Full | Hard | | |
| *Base: Qwen2.5-Coder-7B-Instruct* | | | | | | | | | | |
| Baseline | 91.5 | 84.8 | 82.8 | 71.4 | 49.5 | 19.6 | 41.8 | **20.3** | 34.2 | 55.1 |
| RLAC (Qwen) | **93.3** | **86.0** | **83.9** | 73.0 | **52.2** | **24.3** | 42.3 | 19.6 | 35.2 | **56.6** |
| RLAC (GPT-4o) | 92.1 | 85.4 | 83.9 | **73.2** | 52.2 | 23.6 | **42.5** | 20.3 | **35.5** | 56.5 |

# I   QUALITATIVE CASE STUDIES

This section presents additional qualitative examples illustrating the behavior of the critic before and after adversarial training.

## I.1   FACTUAL TEXT GENERATION CASE STUDY

**Generator output(numbered)**

```
[1] Amina Wadud is a prominent Islamic scholar and theologian based in
    the United States.
[2] She is known for her groundbreaking work in Islamic studies,
    particularly in reinterpreting Islamic texts for modern, gender-
    inclusive contexts.
[3] Wadud has been a leading voice in the movement toward a more
    inclusive and progressive understanding of Islamic law and theology.
[4] She is a professor at the University of Durham and has held academic
    positions at several other institutions.
[5] Wadud is widely recognized for her advocacy of women's religious and
    social rights within Islam.
[6] She is the first woman to lead a Friday prayer in a major American
    mosque.
[7] Her research and writings challenge traditional interpretations of
    Islamic law and promote gender equality.
[8] Wadud has been an influential figure in interfaith dialogue and
    religious pluralism.
```

**Initial Critic Output**

```
reason: Amina Wadud is not a prominent Islamic scholar based in the
    United States.
sentence: 1
error_fact: Amina Wadud is a prominent Islamic scholar based in the
    United States.
```

**Trained Critic Output**

```
reason: The Friday prayer Amina Wadud led took place in a church hall in
    New York, not in a mosque.
sentence: 6
error_fact: Amina Wadud led this Friday prayer in a major American mosque
     in the United States.
```

## I.2    CODE GENERATION CASE STUDY

**Code Question**

```
You are tasked with implementing a function that simulates the behavior
    of a drawing application by managing a canvas. The canvas is
    represented by a 2D grid of pixels, where each pixel can be either
    empty (0) or filled (1). Your function should handle the following
    operations: 1) Resize the canvas to a new width and height, filling
    the new area with empty pixels. 2) Fill the entire canvas with a
    specified color (1 for filled, 0 for empty). 3) Clear the canvas by
    setting all pixels back to empty (0). Implement a single function
    called `manage_canvas` that takes the following parameters: `canvas`
    (a list of lists representing the initial canvas), `operation` (a
    string indicating the operation: 'resize', 'fill', or 'clear'), and `
    params` (a tuple with parameters for the operation). The function
    should return the modified canvas after performing the specified
    operation. The function should not access any external resources.
```

**Generator output**

```python
from typing import List, Tuple, Union

def manage_canvas(
    canvas: List[List[int]],
    operation: str,
    params: Tuple[Union[int, float, bool]]
) -> List[List[int]]:
```

```
    """Manage the canvas based on the specified operation and parameters
    ."""
    if operation == 'resize':
        # params = (new_width, new_height)
        new_width, new_height = params
        new_canvas = [[0] * new_width for _ in range(new_height)]
        return new_canvas

    elif operation == 'fill':
        # params = (fill_value,)
        fill_value, = params
        return [[fill_value] * len(row) for row in canvas]

    elif operation == 'clear':
        return [[0] * len(row) for row in canvas]

    else:
        raise ValueError("Invalid operation")
```

**Initial Critic Output**

```
<think>
1. The function supports a 'resize' operation that changes the canvas to
   (new_width, new_height).
2. Typical boundary cases include an empty canvas and zero or negative
   dimensions.
 </think>
<testcase> CALL: manage_canvas([], 'resize', (2, 2)) </testcase>
```

**Trained Critic Output**

```
 </think>
1. The code does not handle the 'resize' operation correctly. It
   initializes a new canvas with the correct dimensions but does not
   copy the existing canvas's data into the new canvas. This could
   result in a canvas with only empty pixels after resizing.
2. This behavior contradicts the problem description, where 'resize' is
   expected to preserve existing pixels and only fill newly created
   areas with 0, as in a normal drawing application.
 </think>
<testcase> CALL: manage_canvas([[1, 1], [1, 1]], 'resize', (3, 3)) </
   testcase>
```

## J  EXTENDED RELATED WORK

**Outcome-reward RL for reasoning.** RL for LLM has been shown to significantly boost model performance in domains where the success of the final answer can be easily checked (OpenAI et al., 2024; Liang et al., 2025; Team et al., 2025; Lambert et al., 2025). This mostly includes the domains of math (Cobbe et al., 2021; Cui et al., 2025; Luo et al., 2025b; Yu et al., 2025), coding (Jimenez et al., 2024; Pan et al., 2024a; Wei et al., 2025; Luo et al., 2025a) , but can be tricky for other domains like agent decision-making (Pan et al., 2024b; Zhai et al., 2024; Bai et al., 2024) and free-form generations (Min et al., 2023; Zhuge et al., 2024). However, RLAC is designed to relax this requirement so that we can apply RL to more general domains where success cannot be easily verified, such as free-form generations.

