# OpenReview forum: "RLAC: Reinforcement Learning with Adversarial Critic for Free-Form Generation Tasks"
_ICLR.cc/2026/Conference — ICLR 2026 Poster_

### Official Review · Reviewer_7h2p · 2025-10-31

**Soundness:** 3
**Presentation:** 3
**Contribution:** 2
**Rating:** 4
**Confidence:** 3

**Summary:**

Post-training methods for large language models (LLMs) constitute an active area of research. However, reinforcement learning (RL)-based fine tuning is very challenging due to the generative nature of LLM outputs. In particular, the design of efficient reward functions is difficult.

To address this issue, the authors introduce Reinforcement Learning from Dynamic Critic Feedback (RLDCF), a framework that focuses on post-training as an adversarial game between a generator and a critic. In RLDCF, each instruction/prompt provided to the LLM is associated with a set of rubrics representing task-specific requirements that the output should satisfy.  The objective is then to train a generator that maximizes the probability of providing correct outputs.

The critic, modeled as a stochastic policy, aims at providing the worst-case criterion for a given instruction–action pair. The generator is then trained by solving a mini–max optimization problem. Both the generator and critic are implemented as LLMs fine-tuned using a DPO loss.

The proposed approach is evaluated using text and code generation tasks. For factual text generation, experiments are conducted on the Wikipedia Biography Dataset using base generators Qwen3-4B and Qwen3-8B, compared against three baseline methods. For code generation, the authors employ the AceCode-87K-hard subset, with base generators Qwen2.5-Coder-7B-Base and Qwen2.5-Coder-7B-Instruct, also benchmarked against three baselines.

**Strengths:**

Post-training methods for large language models is a crucial task to improve task specific use of generative models and provide robust LLMs.

The presentation of the paper is clear, the problem is well motivated and the overall description of the method is good.

Experimental results demonstrate that RLDCF yields competitive results in both text and code generation quality, highlighting the effectiveness of adversarial critic feedback to finetune.
- In the text generation experiment, RLDCF achieves the same level of factuality as FacTune-FS with fewer verification calls, it also improves KL divergence along epochs with monotonic FactScore gains.
- In the code generation task, the proposed approach outperforms enumerative method  and static reward model method for most benchmarks.

**Weaknesses:**

The results are interesting and promising on the two proposed tasks. However, as the paper is mostly experimental, I would expect more discussion on the choice of the methodological choices. For instance on the way the critic and generator are updated. The influence of K (candidate outputs for each instruction) or N (number of criteria sampled from the critic) should be strong on the results. Even if no theoretical guarantees are provided (which is a probably a very hard question), I would expect more discussions on these "hyperparameters".

In the experiments, it is hard to assess the statistical significance of the results are there is no uncertainty quantification (standard deviation of the metrics for independent runs for instance).

Although adding additional experiments or simulations is not strictly necessary to demonstrate the soundness of RLDCF, the contribution being primarily methodological, its impact would be strengthened if the empirical evaluation, especially for the factual text generation task, were illustrated with a broader variety of performance results.

Minor weakness: the paper would benefit from a careful proofreading to remove typos (Appendix and Tables wrongly referenced for instance).

**Questions:**

In both experiments N and K are fixed. However, these parameters should have a great impact on the results. Can you discuss this ?
These hyperparameters are different in both experiments. Is there a reason for this ?

In the experiments, can you add some qunatification of uncertainty (std over various runs for instance) ?

Is it possible to highlight the performance of RLDCF on other text generationt tasks to support the applicability of the method ? For instance using datasets used in papers associated with the baselines (medical question answering of [Tian et al., 2024] for instance).

Does changing the backbone model for the generator and critic have an influence on the experimental conclusions (factuality level/number of calls, dynamics of the KL divergence along epochs) ?

---

> ### Author Response · Authors · 2025-11-23
>
> Thank you for your constructive suggestions. To address your comments, we (1) clarify the methodological choices and the roles of K and N across different tasks, (2) add uncertainty quantification by running RLDCF with multiple random seeds on the Qwen3-4B 8 Sentences factual text generation task, and (3) add new experiments on a medical QA dataset and with an alternative backbone (LLaMA-3.1-8B-Instruct) to demonstrate robustness across domains and models, in addition to fixing the minor issues you pointed out. **We detail these changes below; if you feel that your concerns are resolved, we would be very grateful if you could consider raising your score.**
>
> 1) Discussion of methodological choices and the role of K and N
>
> For the factual text generation experiment, we set K=10, which matches the configuration used in FactTune-FS (our main baseline). For the code generation task, we use K=8, consistent with the AceCoder setup. Thus, the difference in K is not arbitrary; it reflects the standard experimental settings adopted in the corresponding prior work.
>
> An important practical consideration is that a larger N reduces the number of generator candidates that pass all checks, and during training, we must ensure that the generator still produces a sufficient number of preference pairs for DPO. At the same time, the number of critic-proposed testcases per candidate cannot be too large, as this would inflate verification cost and introduce many redundant checks.
>
> This also explains why the hyperparameter settings differ across tasks: factual verification has a larger and more diverse rubric space (requiring a larger N for exploration), and the biography dataset itself is relatively small; in contrast, code generation uses more deterministic unit tests and contains over 2K problems, so a smaller N already suffices to provide ample preference pairs. We have added the discussion of this section to Appendix D.
>
> 2) Uncertainty quantification
>
> We have run RLDCF with three random seeds on the Qwen3-4B-8sentences factual text generation task.
>
> **Table 4.** Factual text generation on 8-sentence biographies with the Qwen3-4B backbone, with results shown as mean ± standard deviation across three runs.
>
> | **Method**                 | **# Corr ↑**           | **# Incorr ↓**         | **FS ↑**              | **Calls ↓**            |
> |---------------------------|------------------------|------------------------|-----------------------|------------------------|
> | *Qwen3-4B Baseline*       | 19.03 ± 0.41           | 12.05 ± 0.19           | 0.616 ± 0.006         | –                      |
> | *Qwen3-4B FactTune-FS*    | 21.67 ± 0.79           | 5.66 ± 0.48            | 0.793 ± 0.017         | 402,781 ± 52,264       |
> | *Qwen3-4B ArmoRM*         | **22.51 ± 1.00**       | 9.39 ± 0.58            | 0.705 ± 0.013         | –                      |
> | **Qwen3-4B RLDCF (Ours)** | 21.45 ± 0.31           | **4.37 ± 0.48**        | **0.831 ± 0.016**     | **70,667 ± 20,072**    |
>
>
> The results show consistent improvements over the baselines with relatively small variance. These updated results have additionally been included in Appendix D of the revised paper.
>
> 3) Additional text-generation tasks and backbone robustness
>
> We appreciate the suggestion to test RLDCF on a broader set of text-generation tasks and backbones. To assess the effect of changing the backbone LLM, we conducted additional experiments using LLaMA-3.1-8B-Instruct as both the generator and critic. We have plotted the resulting curves and included them in **Appendix E**.
> We observed qualitatively similar behavior: RLDCF consistently increases factual correctness and decreases verification calls, achieving higher factual accuracy in the high-KL regime. We will report these additional results in an appendix to highlight that our conclusions do not depend on a specific backbone choice.
> In addition to Wikipedia biographies, we have experimented on medical QA, as used in prior work. Given the tight schedule, we still need time to finalize the results.
>
> 4) Minor issues
>
> We will carefully proofread the paper and fix the typos and incorrect appendix/table references.

---

> > ### Author Response · Authors · 2025-11-27
> >
> > We evaluated our method on a MedicalQA dataset. Since the authors of FactTune-FS did not release their MedicalQA dataset,
> > we constructed our own by randomly sampling 170 common medical conditions from Wikipedia topics, using 120 for training and 50 for testing.
> >
> > **Table 5.** Factual text generation on 8-sentence MedicalQA questions with the Qwen3-4B backbone.
> >
> > | **Method**                 | **# Corr ↑** | **# Incorr ↓** | **FS ↑** | **Calls ↓** |
> > |---------------------------|--------------|----------------|----------|-------------|
> > | *Qwen3-4B Baseline*       | 34.0         | 3.38           | 0.909    | –           |
> > | *Qwen3-4B FactTune-FS*    | 34.2         | 1.99           | 0.945    | 524,329     |
> > | *Qwen3-4B ArmoRM*         | 35.1         | 2.36           | 0.937    | –           |
> > | **Qwen3-4B RLDCF (Ours)** | **35.2**     | **1.67**       | **0.955**| **76,800**  |
> >
> > The results follow the same trend observed on the biography dataset, demonstrating the generality and robustness of our approach. These updated results have additionally been included in Appendix F of the revised paper.

---

### Official Review · Reviewer_DxZj · 2025-11-01

**Soundness:** 2
**Presentation:** 4
**Contribution:** 3
**Rating:** 4
**Confidence:** 4

**Summary:**

The paper tackles the important issue of the most likely failure case of generated free-form text by introducing a novel adversarial learning scheme for training a generator and critic. The generator outputs multiple candidate solutions and the critic, given a candidate, finds the case to most likely fail given the generation. They employ a DPO loss for both the generator and critic. They have consistent results on factual text generation and code generation showing improved performance over baseline methods.

**Strengths:**

-The paper is well-written and easy to follow

-Paper outperforms baselines methods in both factual text generation and code generation

-The proposed approach is novel, applying adversarial learning to the important, significant problem of identifying failure cases in free-form generation

-I find the use of DPO fairly interesting and novel

**Weaknesses:**

-For fact verification, I do not necessarily see the need for a critic to specify which fact the check. Can one not separate out all the facts in the generated responses, either programmatically or with an LLM, and then run each one through the fact verifier? I understand from later on in the paper the verification is costly, but verifying each fact would provide more accurate rewards for the generator, correct? For code generation, I understand this simplification is not possible because the test cases cannot be parsed from the generated code.

-If I understand correctly, given a candidate answer a_i to s, the reward that is used to fine-tune the generator is sparse. Therefore, if even one fact is incorrect or one test case is not correct, the whole candidate output is assigned a reward of 0, correct? If so, in the case of code generation, that may be too strict because realistically code is regularly updated to handle unseen test cases, not marked as entirely wrong. Also for fact generation, if one fact is wrong, the sparse reward does not help the generator learn which fact was wrong. I understand the issue may be mitigated by the sampling multiple candidates and using DPO loss, so similar candidate solutions with different rewards can help the generator learn with finer-grained feedback. However, how many times, especially in the beginning of training, do you get a group of candidates with reward = 0 and reward = 1 to provide some distinction to the generator?

-I’m not too familiar with FactScore, so I am not sure what is the bottleneck cost for verification mentioned in L249. But for code generation, I do not see why number of test cases executed is a good metric for efficiency analysis. Test cases, especially for the ones provided in the used datasets, can be quickly executed.

**Questions:**

See Weakness 1 and 2

Minor Suggestions/Weaknesses:
-L86 has two empty parentheses
-Missing reference in L248

---

> ### Author Response · Authors · 2025-11-23
>
> Thank you for your helpful comments. To address your concerns, we clarify (1) why exhaustive fact extraction corresponds exactly to the Factune-FS baseline and why it becomes infeasible for long texts, (2) why our binary verifier still provides an effective and stable learning signal and outperforms denser reward schemes, and (3) how our two experimental settings correspond to fundamentally different verification bottlenecks, thereby justifying the use of executed test cases as an efficiency metric only in the factual-setting experiments. **We provide detailed explanations below; if you feel that your concerns are resolved, we would be very grateful if you could consider raising your score.**
>
> > “Why not extract all facts and verify each one individually?”
>
> First, we would like to clarify how FactScore works: the text is decomposed into minimal atomic facts by an LLM, and each fact is then verified independently. Specifically, FactScore retrieves the five most relevant paragraphs from the corresponding Wikipedia page, and the LLM determines whether the atomic fact is correct based on the retrieved evidence. The final score is aggregated from these per-fact judgments. Therefore, **what you described corresponds exactly to the Factune-FS baseline used in our experiments.**
> In FactScore, the most expensive component is the independent verification of each fact, which is defined as the number of verification calls. Verifying a single atomic fact requires approximately **11.43 Tflops**, and verifying an 8-sentence biography therefore costs about $0.03–$0.05. According to our experimental results, a 4-sentence biography contains about 17 atomic facts, whereas an 8-sentence biography contains about 30 atomic facts. Thus, the verification cost grows linearly with text length and becomes prohibitively high as the text becomes longer.
>
> Moreover, exhaustively verifying all facts does not necessarily yield a more accurate reward signal in practice. The process of extracting “all facts” using programmatic tools or LLMs can itself introduce noise. Here is an illustrative example. For the sentence
>
> “Aaron Kwok is a renowned Hong Kong singer and actor known for his versatile talent in both music and film.”
>
> A decomposition may yield atomic facts such as:
>
> (f₁) Aaron Kwok is a singer.
> (f₂) Aaron Kwok is an actor.
> (f₃) Aaron Kwok is a Hong Kong singer.
> (f₄) Aaron Kwok is a Hong Kong actor.
> (f₅) Aaron Kwok is a renowned Hong Kong singer.
> (f₆) Aaron Kwok is a renowned Hong Kong singer and actor.
> (f₇) Aaron Kwok has versatile talent in both music and film..
>
> Several of these atomic facts are overlapping or near-duplicates. Nevertheless, each of them is verified independently and contributes separately to the aggregated score, making the resulting reward noisy.
>
> In contrast, RLDCF does not rely on such exhaustive fact decomposition. It only verifies the failure modes proposed by the critic, which results in a much cleaner and more stable signal.
> Our contribution is precisely that RLDCF achieves FactScore-level factual accuracy improvements while requiring dramatically fewer validator calls in settings with expensive validators. The results further show that when scaling from 4-sentence biographies to longer 8-sentence ones, RLDCF’s verification-cost advantage over Factune-FS widens, demonstrating the scalability of our method.

---

> > ### Author Response · Authors · 2025-11-23
> >
> > > “Is the reward too sparse?”
> >
> > Before adopting a binary verifier, we first implemented a denser scoring scheme where the reward reflected how many critic-proposed rubrics detected an error (e.g., assigning different reward levels to 1/4, 2/4, 3/4, and 4/4 failures). However, this variant consistently performed worse. The reason is that mid-range distinctions such as “2 out of 4 failures” vs. “3 out of 4 failures” are dominated by retrieval randomness rather than reflecting meaningful differences in model quality. These unstable intermediate scores directly introduce noise into the preference pairs used for DPO training.
> >
> > In contrast, binarizing the verifier leads to cleaner and more reliable supervision: only clear-cut failures are marked as such, which substantially reduces label noise during preference construction. Despite this binarization, the generator still receives sufficiently informative feedback because we sample multiple diverse candidates (temperature = 1.0). At the beginning of training, the proportion of effective 0–1 preference pairs is already substantial, **54%** / **83%** for Qwen3-4B and Qwen3-8B in factual text generation, and **48.96%** / **67.49%** for the Qwen2.5-Coder-Base and Qwen2.5-Coder-Instruct in code generation.
> >
> > Of course, we agree that a dense reward signal can in principle, be beneficial. The key challenge is how to construct such fine-grained signals without amplifying retrieval noise or extraction artifacts. Designing robust, high-resolution reward schemes for free-form generation remains an interesting open research question, and we view our binary verifier as a practical, stable step in that direction.
> >
> > > “Why is the number of executed test cases an efficiency metric? Testcases are cheap.”
> >
> > We apologize for the misunderstanding here. In our paper, the two experiments are designed with different goals in mind, corresponding to two distinct types of verification bottlenecks that arise in practice. We have clarified this distinction explicitly in Section 4.2.1 (“Settings”) now.
> > The first type, exemplified by FactScore, is exhaustively verifiable but incurs extremely high verification costs. The second type, as in code generation, involves tasks where the space of potential test cases cannot be enumerated.
> > Therefore, verification cost is not the primary metric in the code-generation study. Instead, our focus is on improvements over standard code benchmarks, since the limiting factor there is not verification scalability but the inability to exhaustively validate all possible test cases.

---

> > > ### Comment · Reviewer_DxZj · 2025-11-23
> > >
> > > Thank you for your detailed response.
> > > 1. Thank you for clarifying FactScore and why the baseline is unscalable. Also, the Aaron Kwok example is a good illustration of why you cannot programmatically parse it, making it much clearer to me why an LLM is needed. Consider putting this example in the main body or the Appendix because I feel like it makes the motivation clearer.
> > > 2. I see. Yes, suboptimal dense reward structures can lead to unstable learning and would be a good avenue for future work.
> > > 3. Thank you for the clarification
> > >
> > > I am happy to raise my score.

---

### Official Review · Reviewer_G4q7 · 2025-11-11

**Soundness:** 2
**Presentation:** 4
**Contribution:** 3
**Rating:** 4
**Confidence:** 4

**Summary:**

This paper tackles a real pain point in training large language models: how do you optimize for tasks where outputs need to satisfy tons of different criteria, and you can't possibly check them all? The authors propose RLDCF, which basically turns training into a game between two models. A generator tries to produce good outputs, while a critic tries to catch the generator's mistakes by proposing specific ways it might fail. An external validator then checks if the critic found a real error. The critic gets better at spotting weaknesses, and the generator gets better at avoiding them. The idea is neat: instead of exhaustively checking every possible criterion or relying on a static reward model that can be gamed, you dynamically focus on the most likely failure points.

The paper suggesr promising results. For biography generation, they hit a FactScore of 0.889 while doing 5.7 times less verification work than existing methods. For code generation, they claim the best scores despite using only 9% of the training data. However, a potential problem in code generation experiments suggest circular logic: they essentially created "reference solutions" using the same model family they're training. Then the model could just be learning what Qwen is already able to do. Qwen-7B-Instruct training shows limited improvement over the base model, could be well within variance, and the authors didn't provide much details on what or how the model is improved.

The core idea of adversarial training idea isn't particularly novel, similar approaches have appeared in recent work on generative verifiers. The biography experiments are more solid and the overall problem they're solving matters. But between the circular validation issues, limited novelty, and some unfair experimental comparisons, this feels like a decent idea that needs another round of work with more solid experiments and applications that can justify sufficient contribution to the area.

**Strengths:**

- Important and natural problems to tackle for rubric based reward modeling and RL training for LLM post training.
- Theoretical formulation is solid.
- Strong factual text generation results.
- Good ablation studies.
- 4 to 8 sentence comparison shows the method scales well to complexity
- Presentation is clear

**Weaknesses:**

- I have doubts about code experiment set up as mentioned above
- Lack of theoretical or empirical analysis into the method and experiment results
- Could benefit from more analysis & learnings and more experiments.

**Questions:**

I'm open to change my score if authors can provide sufficient justification or insights in both of the experiments, especially the coding one

---

> ### Author Response · Authors · 2025-11-27
>
> We thank the reviewer for the thoughtful comments on our verification protocol, the relation to generative verifiers, and the behavior of the evolving critic. In the revision, we (1) clarify the verification process and add a cross-family validation experiment using GPT-4o as the reference solution, (2) more clearly distinguish RLDCF from static generative verifiers, and (3) provide a case study analyzing how the critic evolves during training. **We address your concerns below, and would appreciate it if you would be willing to raise your score if you find your concerns addressed.**
>
> 1) Clarification of the code-generation verification setup.
>
> We emphasize that **neither the generator nor the critic ever has access to the reference solution**, its outputs, or any information derived from it. The external validator acts only as a black-box function: given a critic-proposed testcase, it executes the same input on (i) the generator’s output and (ii) the reference solution, and returns a **single binary bit** indicating agreement or disagreement (also checks whether the test case triggers a valid output from the reference solution rather than an immediate error, to prevent exploitation).
> No information from the reference solution is ever exposed to either policy.
>
> To further rule out any family-specific leakage, we ran a cross-family validation experiment, using GPT-4o to generate the reference solution.
>
> **Table 6.** Results for HumanEval, MBPP, BigCodeBench Complete and Instruct (BCB-C, BCB-I), and LiveCodeBench (LCB). Using a different model (GPT-4o) to generate reference solutions, the results show that **RLDCF** still achieves a similar level of average performance improvement across all benchmarks.
>
> _Base model: Qwen2.5-Coder-7B-Instruct_
>
> | **Method**              | **HumanEval Base** | **HumanEval Plus** | **MBPP Base** | **MBPP Plus** | **BCB-C Full** | **BCB-C Hard** | **BCB-I Full** | **BCB-I Hard** | **LCB**        | **Average**      |
> |-------------------------|--------------------|--------------------|---------------|---------------|----------------|----------------|----------------|----------------|----------------|------------------|
> | Baseline                | 91.5               | 84.8               | 82.8          | 71.4          | 49.5           | 19.6           | 41.8           | **20.3**       | 34.2           | 55.1             |
> | **RLDCF (Qwen)**        | **93.3**           | **86.0**           | **83.9**      | 73.0          | **52.2**       | **24.3**       | 42.3           | 19.6           | 35.2           | **56.6**         |
> | RLDCF (GPT-4o refs)     | 92.1               | 85.4               | **83.9**      | **73.2**      | **52.2**       | 23.6           | **42.5**       | **20.3**       | **35.5**       | 56.5             |
>
> The results remained consistent with the original results, indicating that RLDCF is not exploiting any overlap between the reference solution and the generator backbone, and that the improvements come from the adversarial training procedure itself rather than copying from Qwen.
>
>
> The relatively modest improvement observed on Qwen-7B-Instruct is expected.
>
> As described in the original AceCoder paper and its reference implementation, this split is constructed by first evaluating two 7B/8B–scale base instruct models, Qwen2.5-Coder-7B-Instruct and LLaMA-3-8B-Instruct, on the full AceCoder-87K dataset. For each problem, AceCoder computes the average pass rate of these two models and then retains only those problems whose average pass rate is at or below the 50th percentile of this distribution. In other words, “hard” is defined relative to these two base models via a median-based threshold, so the AceCoder-87K-hard subset contains all problems whose difficulty is at least median-level for these models rather than only the most extreme tail.
>
> This construction provides substantial headroom for a weaker base model, thus RL on this split can yield large gains. For a stronger instruct model such as Qwen2.5-Coder-7B-Instruct, however, the starting performance on this “hard” split is already much higher: many of the median-or-below problems are no longer truly difficult, and further improvement would require training on an even more challenging and diverse subset. Put differently, AceCoder-87K-hard may not cover a sufficiently rich set of genuinely difficult patterns for a strong instruct model, which naturally limits the observable improvement size in our setting. In addition, our setup uses only 2k training problems and far fewer optimization steps than AceCoder-RM, which trains on over 163,840 samples. We will clarify these details in the revision.

---

> ### Author Response · Authors · 2025-11-27
>
> 2) RLDCF is fundamentally different from generative verifiers.
>
> Existing verifier-based approaches treat the judge as a static evaluator: the verifier outputs fixed labels or explanations and does not change its behavior as the generator improves. These methods are therefore functionally equivalent to reward models whose outputs remain fixed throughout training.
> In contrast, RLDCF trains the critic as a stochastic policy in an adversarial loop.
>  The critic continually adapts its proposals toward increasingly challenging failure modes as the generator evolves, effectively shaping the difficulty curriculum. To the best of our knowledge, no prior work trains a generative verifier dynamically in such a closed-loop setting.
> Furthermore, generative verifiers typically give the reward, while RLDCF never uses the critic to assign scores; the reward signal comes entirely from the external validator’s binary correctness flag. While a design where the evaluator provides a reward estimate is possible, this eliminates reward hacking and removes dependencies on evaluator phrasing or calibration, a known issue in reward-model-based RL.
>
> 3) Critic behavior evolution.
>
> To provide the requested insight into how the critic improves, we analyzed its proposed rubrics throughout training. The example of the task description and the model outputs is included in Appendix H.
>
> Factual Text Generation (Appendix H.1)
>
> Consider the following paragraph about Amina Wadud generated by the initial generator (sentences are numbered for clarity):
> ```
> [1] Amina Wadud is a prominent Islamic scholar and theologian based in the United States.
> [2] She is known for her groundbreaking work in Islamic studies, particularly in reinterpreting Islamic texts for modern, gender-inclusive contexts.
> [3] Wadud has been a leading voice in the movement toward a more inclusive and progressive understanding of Islamic law and theology.
> [4] She is a professor at the University of Durham and has held academic positions at several other institutions.
> [5] Wadud is widely recognized for her advocacy of women's religious and social rights within Islam.
> [6] She is the first woman to lead a Friday prayer in a major American mosque.
> [7] Her research and writings challenge traditional interpretations of Islamic law and promote gender equality.
> [8] Wadud has been an influential figure in interfaith dialogue and religious pluralism.
>  ```
> One of the critic’s initial sampled outputs was:
> ```
> reason: Amina Wadud is not a prominent Islamic scholar based in the United States.
> sentence: 1
> error_fact: Amina Wadud is a prominent Islamic scholar based in the United States.
> ```
> This response simply negates an actually correct statement in sentence [1], without decomposing the sentence or identifying any real factual inconsistency. It reflects a typical early-stage behavior: the critic mislabels true facts as errors and fails to localize the genuine mistake in the biography.
> After training, the later-stage critic, when shown the same paragraph, instead proposes:
> ```
> reason: The Friday prayer Amina Wadud led took place in a church hall in New York, not in a mosque.
> sentence: 6
> error_fact:.Amina Wadud led this Friday prayer in a major American mosque in the United States.
> ```
> In this later stage, the critic learns to identify factual errors and to correctly produce rubrics that can be used for external verification. Meanwhile, the generator gradually reduces incorrect facts and increases the number of correct ones.
> As training progresses, the critic can shift its attention and continually discover new factual errors that emerge from the generator’s updated outputs.
> As the critic becomes more precise in identifying failure modes, the generator is in turn guided toward systematically reducing these errors, leading to the monotonic improvements in factual accuracy and code correctness observed in our experiments.

---

> > ### Author Response · Authors · 2025-11-27
> >
> > Code generation (Appendix G.2)
> >
> > We observe a similar progression in the code domain. Consider a prompt asking the model to implement a canvas management function (see Appendix G.2 code generation prompt),
> >
> > An initial generator output is:
> > ```
> > from typing import List, Tuple, Union
> >
> > def manage_canvas(
> >     canvas: List[List[int]],
> >     operation: str,
> >     params: Tuple[Union[int, float, bool]]
> > ) -> List[List[int]]:
> >     """Manage the canvas based on the specified operation and parameters."""
> >     if operation == 'resize':
> >         # params = (new_width, new_height)
> >         new_width, new_height = params
> >         new_canvas = [[0] * new_width for _ in range(new_height)]
> >         return new_canvas
> >
> >     elif operation == 'fill':
> >         # params = (fill_value,)
> >         fill_value, = params
> >         return [[fill_value] * len(row) for row in canvas]
> >
> >     elif operation == 'clear':
> >         return [[0] * len(row) for row in canvas]
> >
> >     else:
> >         raise ValueError("Invalid operation")
> > ```
> >
> > At early stages of training, the critic mostly focuses on boundary conditions and trivial inputs, for example, proposing testcases such as:
> > ```
> > <think>
> > 1. The function supports a 'resize' operation that changes the canvas to (new_width, new_height).
> > 2. Typical boundary cases include an empty canvas and zero or negative dimensions.
> >  </think>
> > <testcase> CALL: manage_canvas([], 'resize', (2, 2)) </testcase>
> > ```
> > Such testcases are often either uninformative (e.g., not exposing a real semantic error) or occasionally non-runnable, and they do not reflect the true mismatch between the specification and the implementation. After adversarial training, the critic begins to target the semantic gap between the intended behavior and the current implementation.
> > A later-stage critic sample is:
> > ```
> >  </think>
> > 1. The code does not handle the 'resize' operation correctly. It initializes a new canvas with the correct dimensions but does not copy the existing canvas's data into the new canvas. This could result in a canvas with only empty pixels after resizing.
> > 2. This behavior contradicts the problem description, where 'resize' is expected to preserve existing pixels and only fill newly created areas with 0, as in a normal drawing application.
> >  </think>
> > <testcase> CALL: manage_canvas([[1, 1], [1, 1]], 'resize', (3, 3)) </testcase>
> > ```
> > Here, the critic (i) constructs a testcase that preserves non-empty initial content, (ii) explicitly captures the discrepancy between the expected and implemented resize semantics, and (iii) produces a runnable call that reliably triggers the failure. As training progresses, such testcases become more diverse and better aligned with the task semantics, illustrating that the critic is not merely checking input validity but actively searching for behavioral weaknesses in the generator. Overall, these examples demonstrate that the critic in RLDCF evolves from noisy, coarse heuristics to targeted, high-value probes in both factual and code settings, supporting our claim that the critic is genuinely learning non-trivial structure rather than acting as a static generative verifier.

---

### Author Response · Authors · 2025-12-01
**Summary of Rebuttal and Responses to Reviewer Concerns**

Dear Area Chair,

Thank you for overseeing the review process of our submission. In response to the reviewers’ concerns, we (i) clarified the code-generation setup and ruled out circularity via a new cross-family GPT-4o reference experiment, (ii) distinguished RLDCF from static generative verifiers and illustrated how the critic meaningfully evolves over training, and (iii) discussed the methodological roles of K and N (iv) strengthening our empirical evidence with multi-seed runs, additional MedicalQA experiments, and LLaMA-3.1-8B-Instruct backbones. These changes resolve the reviewers’ concern and Reviewer DxZj explicitly raise their score to 8 (poster). Below we summarize the key points in more detail.


## Main concerns raised and addressed during rebuttal


> Code-generation setup and potential circularity

We clarified the operating mechanism of the external validator and added a cross-family experiment. Specifically, Appendix G now includes a cross-family setting with GPT-4o, where RLDCF still achieves similar improvements. We also explained that the dataset split and the limited number of optimization steps constrain the magnitude of performance gains observed for Qwen2.5-Coder-7B-Instruct.


> Relationship to generative verifiers and explanation of model improvements

We clarified the conceptual distinction between the adversarial critic and generative verifiers, and added case studies for factual text and code in Appendices H.1 and H.2, showing how the critic evolves from noisy, coarse rubrics into more targeted tests that help the generator improve.

> Necessity of the critic and reward sparsity

We clarified that the “verify all facts” setting corresponds exactly to the FactTune-FS baseline, and used the Aaron Kwok example to further illustrate how such exhaustive decomposition introduces many overlapping or near-duplicate facts, thereby adding noise. We explained why a more “dense” reward scheme performs worse, and provided empirical evidence that even early in training, a substantial fraction of candidate sets already yields highly informative 0–1 preference pairs.


> Methodological choices for K and N

We clarified that the choice of K follows the standard configurations used in prior work, while N controls the trade-off between exploring the rubric/test space and retaining enough successful candidates for DPO. We added a discussion in Appendix C explaining why the value of N differs across tasks.

> Uncertainty quantification

We added multi-seed experiments on the Qwen3-4B 8-sentence biography setting and reported mean ± standard deviation in Appendix D Table 4. The results show that RLDCF consistently outperforms the baselines with relatively small variance, addressing concerns about the statistical significance of our improvements.

> Robustness across tasks and backbones

We evaluated RLDCF with LLaMA-3.1-8B-Instruct as both generator and critic (Appendix E), observing similar trends in factual accuracy and verification calls as with Qwen backbones. We also added a MedicalQA experiment(Table 5, Appendix F), where RLDCF again achieves the best FactScore with far fewer validator calls than FactTune-FS and ArmoRM, supporting robustness across datasets and domains.


## Summary of reviewer responses after rebuttal

- Reviewer DxZj (initial score: 4): Confirmed that the rebuttal addressed all concerns and raised the score to 8.
- Reviewer 7h2p (initial score: 4): Did not post a follow-up before the freeze, but concerns have now been addressed.
- Reviewer G4q7 (initial score: 4): Did not post a follow-up before the freeze, but concerns have now been addressed.

We hope this summary is helpful for your decision-making. Thank you again for your time and consideration. Please let us know if you have any questions.

---

### Meta-Review · Area_Chair_3AYx · 2026-01-04

**Summary:**

This paper proposes Reinforcement Learning from Dynamic Critic Feedback (RLDCF), a framework for scalable RL post-training on free-form generation tasks where exhaustive rubric verification is impractical. The approach trains a generator and a dynamic LLM critic adversarially, using an external validator to focus verification on likely failure modes. Experiments on factual text and code generation show improved quality with fewer verification calls.

**Reviewer Concerns:**

Reviewers raised concerns about (i) potential circularity in the code-generation setup due to same family reference solutions, (ii) limited novelty relative to prior generative-verifier and adversarial-critic approaches, (iii) experiment comparison fairness and clarity, and (iv) lack of uncertainty quantification.

The rebuttal addressed several of these points with new evidence. The authors added a GPT-4o reference experiment to mitigate same-family leakage issue, made a clearer conceptual distinction between dynamic critics and static generative verifiers, added concrete case studies to illustrate critic evolution, and added multi-seed results (with variance). The authors also added evaluations on MedicalQA and alternative backbones. These steps partially resolve the circularity and robustness concerns.

However, important limitations remain. The code-generation protocol is complex and the reported gains are modest, leaving some doubts about whether improvements exceed variance across settings. Novelty remains incremental. Uncertainty quantification is still limited. Under a normal review cycle with full discussion, some of these issues can lead to a rejection or at least requires further clarification.

Given this year's disrupted review process, reviewers did not have a full opportunity to assess the new evidence added in rebuttal. In that context, the paper is more of a borderline.

**Reviewer Scores:**

Reviewer DxZj: Increased from 4 to 8 after rebuttal (according to the authors).

Reviewer G4q7: Likely chang to 5-6. concerns about novelty and code-evaluation robustness are partially addressed but not fully resolved.

Reviewer 7h2p: Likely chang to 5-6; methodology and uncertainty concerns improved.

---

### Decision · Program_Chairs · 2026-01-26

Accept (Poster)